# Evaluation of Bioactive Compounds and Antioxidant and Cytotoxic Effects of Oil and Pulp without Açaí Fat (*Euterpe oleracea*) Obtained by Supercritical Extraction

**DOI:** 10.3390/foods13172819

**Published:** 2024-09-05

**Authors:** Letícia Maria Martins Siqueira, Ana Luiza de Barros Souza Campos, Flávia Cristina Seabra Pires, Maria Caroline Rodrigues Ferreira, Ana Paula de Souza e Silva, Eduardo Gama Ortiz Menezes, Ingryd Nayara de Farias Ramos, André Salim Khayat, José de Arimateia Rodrigues do Rêgo, Raul Nunes de Carvalho Junior

**Affiliations:** 1PRODERNA (Graduate Program in Engineering of Natural Resources of the Amazon), LABEX (Extraction Laboratory), LABTECS (Supercritical Technology Laboratory), ITEC (Institute of Technology), UFPA (Federal University of Pará), Augusto Corrêa Street S/N, Guamá, Belém 66075-900, PA, Brazil; 2LABEX (Extraction Laboratory), FEQ (Faculty of Chemical Engineering), ITEC (Institute of Technology), UFPA (Federal University of Pará), Rua Augusto Corrêa S/N, Guamá, Belém 66075-900, PA, Brazil; analu0751@gmail.com; 3LABTECS (Supercritical Technology Laboratory), Science and Technology Park, Perimeter Avenue of Science, km 01, Guamá, Belém 66075-750, PA, Brazil; flaviapiress@gmail.com; 4LABEX (Extraction Laboratory), LABTECS (Supercritical Technology Laboratory), PPGCTA (Graduate Program in Food Science and Technology), ITEC (Institute of Technology), UFPA (Federal University of Pará), Augusto Correa Street S/N, Guamá, Belém 66075-900, PA, Brazil; carolinerof@gmail.com (M.C.R.F.); anapaula-eng@hotmail.com (A.P.d.S.e.S.); 5Federal Institute of Education, Science and Technology of Rondônia, IFRO, Porto Velho 78900-000, RO, Brazil; ortizegom@hotmail.com; 6NPO (Center for Research in Oncology), PPGOCM (Graduate Program in Oncology and Medical Sciences, ICB (Institute of Biological Sciences), UFPA (Federal University of Pará), Augusto Corrêa Street S/N, Guamá, Belém 665-07900, PA, Brazil; ingryd.ramos@icb.ufpa.br; 7NPO (Oncology Research Center), ICB (Institute of Biological Sciences), UFPA (Federal University of Pará), Augusto Corrêa Street S/N, Guamá, Belém 66075-900, PA, Brazil; andrek@ufpa.br; 8LABA (Laboratory of Amazonian Biofilms), PPGCMA (Graduate Program in Sciences and Environment), ICEN (Institute of Exact and Natural Sciences), UFPA (Federal University of Pará), Augusto Corrêa Street S/.CON, Guamá, Belém 665-07900, PA, Brazil; jr2rego@gmail.com; 9LABEX (Extraction Laboratory), LABTECS (Supercritical Technology Laboratory), FEA (Faculty of Food Engineering), ITEC (Institute of Technology), UFPA (Federal University of Pará), Augusto Corrêa Street S/N, Guamá, Belém 665-07900, PA, Brazil

**Keywords:** supercritical CO_2_, oil extraction, green process, anthocyanins, carotenoids, fat-free açaí

## Abstract

For bioactivity studies, it is necessary to use products with a high degree of purity, which may influence the cytotoxic effects. Supercritical technology presents itself as an alternative to obtain these products. Therefore, the objective of this work was to obtain the bioactive compounds of oil and pulp of açaí fat-free supercritical technology and evaluate the cytotoxicity of products in MRC-5 and VERO cells in vitro. The açaí pulp was subjected to extraction with supercritical CO_2_ to obtain the oil and pulp without fat, under conditions of 323.15 K at 35 MPa, 333.15 K at 42 MPa, and 343.15 K at 49 MPa. The largest yields (51.74%), carotenoids (277.09 µg/g), DPPH (2.55 μmol TE/g), ABTS (2.60 μmol TE/g), and FRAP (15.25 μm of SF/g) of oil and ABTS (644.23 μmol TE/g) of pulp without fat were found in the condition 343.15 K at 49 MPa. The highest levels of compounds phenolics (150.20 mg GAE/g), DPPH (414.99 μmol TE/g), and FRAP (746.2 μm SF/g) of the pulp without fat were found in the condition of 323.15 K to 35 MPa. The fat-free pulp presented high levels of anthocyanins without significant variation in cytotoxicity. The developed process was efficient in obtaining oil rich in carotenoids, and the supercritical technology elucidated an efficient way to obtain açaí fat-free pulp.

## 1. Introduction

The growth of the açaí production chain (*Euterpe oleracea*) and its byproducts raised the status of this local product to a global product [1]. This fruit has been explored by the market not only through the food industry in juices, ice cream, sweets, and food supplements but also in formulations for high-performance products from the pharmaceutical, cosmetics, and raw materials industries [2]. This is due to its great nutritional value, being rich in lipids, vitamins, and proteins, its high concentration of fiber, voluminous amount of monounsaturated and polyunsaturated fatty acids and minerals, and providing an excellent source of antioxidants such as phenolic compounds, anthocyanins, and carotenoids [3,4,5].

High levels of anthocyanins and other flavonoids, a class of phenolic compounds, are associated with the health benefits of açaí against various diseases related to oxidative stress [6], as well as antioxidant [7], anti-inflammatory [8], and carcinogenic properties [9]. The phenolic compounds of açaí are associated with health-promoting properties and the prevention of different degenerative diseases [10]. In the compounds obtained in the pulp of açaí, Da Silveira and Godoy [11] identified the following main phenolic compounds: anthocyanins, 3.4 dihydroxybenzoic acids, p-dihydroxybenzoic acid, vanillic acid, caffeic acid, syringic acid, p-cumaric acid, isoorientin, orientin, and ferulic acid. Anthocyanins are one of the main bioactive compounds of açaí pulp and belong to the group of phenolic compounds, and their properties are related to health, namely the protection of cardiovascular diseases, anti-inflammatory effects, and neuroprotective and antiproliferative effects [12,13,14,15]. As the main anthocyanins in açaí, Garzon et al. [7] identified cyanidin 3-O-glycoside, cyanidin 3-O-rutinoside, peonidin 3-O-glycoside, and peonidin 3-O-rutinoside.

Another class that stands out is carotenoids, which recent work suggests that the beneficial actions of these compounds are attributed to antioxidant mechanisms that promote health [16]. A study developed by Torma et al. [17] identified lutein, zeaxanthin, α- carotene, and β-carotene as the main carotenoids in açaí, and the β-carotene corresponded to 47.16% of the total carotenoid content. Beta-carotene is the most studied among carotenoids because it has the highest provitamin A activity and is used as an antioxidant to reduce cellular or tissue damage. It has been reported to have a protective effect against cardiovascular diseases [18,19].

However, recent studies have noted that these compounds also include pro-oxidant mechanisms, damaging cells and tissues [20]. A high intake of β-carotene, for example, is contraindicated for smokers due to increased risk of lung and stomach cancer [21], and it may cause carotenemia [22]. In addition, anthocyanin at high concentrations may present cytotoxicity and is associated with increased mortality rates of brine shrimp cells [23]. Thus, it is essential to verify the cytotoxicity, that is, the ability of these compounds to affect cell viability to enable the application in different industrial branches. In this context, the cell lines MRC-5, originating from the human lung tissues of a 14-week-old fetal man, and VERO, from the kidney of an African green monkey, are sensitive cells and widely explored in various branches of research, including to analyze cytotoxicity and cytoprotection in extracts [24,25,26,27,28].

For cytotoxicity studies, it is necessary to use products with high purity that are free of chemical contaminants that may influence the cytotoxic effects. Supercritical technology presents itself as an alternative to obtain products with a high degree of purity, free of organic waste, and that are environmentally “green”, because it is a technique that uses CO_2_ as a solvent, which is inert and does not offer risks of secondary reactions, such as oxidations, reductions, hydrolysis, and chemical degradation. It is safe, nonexplosive, nonpolluting, and nontoxic [29,30]. The advantages of supercritical fluid extraction technology make this technique recognized as a promising procedure for applications in the food, pharmaceutical, and cosmetic industries due to its high selectivity, shorter extraction time compared with conventional techniques, and the fact that it does not use toxic organic solvents in the extracts [31].

For this reason, extraction with supercritical CO_2_ (CO_2_-SFE) has been used, as it is a sustainable and selective technology that promotes innovative, 100% natural products that contribute to the preservation of the environment and instigate the bioeconomy; however, there is still no marketing of açaí products using this technology. This is because few studies portray the extraction involving supercritical fluids to obtain oils and pulps without açaí fat [32,33], although the development of this extraction is a viable process for obtaining extracts of high quality [34]. The açaí fruit holds significant socioeconomic relevance for the Amazon region due to its enormous potential for full utilization. However, the most profitable and widely used form of exploitation is the production and commercialization of açaí pulp [35]. The market has been showing steady growth, highlighting the importance of developing new value-added products and the need to invest in sustainable processes for the region’s development.

Therefore, the objective of this work is to evaluate the cytotoxicity in MRC-5 and VERO cells in vitro and to determine the main bioactive compounds and antioxidant effects of the obtained products: pulp without fat and açaí oil (*Euterpe oleracea*) by supercritical CO_2_.

## 2. Materials and Methods

### 2.1. Raw Material and Sample Preparation

The fruits of the açaí plant (*Euterpe oleracea*) were collected in the summer harvest, the driest season. The fruits exhibit uniform ripening and better quality in the municipality of Igarapé-Miri/PA/Brazil (1°58′37″ S and 48°57′34″ W) (Sisgen: ABFE510). The raw material was washed with drinking water to remove the impurities and immersed in tanks with water at 323.15 K for softening the mesocarp and endocarp for 900 s. After this treatment, the fruits were pulped with the aid of mechanical pulping. The pulp was packed in plastic bags coated with aluminum foil and transported in refrigerated thermal boxes (278.15 ± 274.15) K. Then, the pulps were added to stainless steel trays and frozen to 255.15 ± 274.15 K. After freezing, the pulp was lyophilized in a freeze dryer (model: L101, LIOTOP, São Carlos, SP, Brazil) under pressure conditions of 1.83 MPa and condenser temperature of 219.15 K for 172,800 s. The lyophilized pulp was packed in vacuum plastic bags coated with aluminum foil and stored under freezing (255.15 ± 274.15) K until extraction.

### 2.2. Characterization of the Lyophilized Pulp

#### 2.2.1. Average Particle Diameter

The particle distribution of the lyophilized sample was determined with the aid of a series of Tyler sieves (W.S. Tyler, Mentor, OH, USA) of 12 to 28 meshes, coupled to a magnetic stirrer (Bertel, model AGMAGB, São Bernardo do Campo, SP, Brazil). The procedure was performed in batches of 0.02 kg, during a period of 1200 s for each batch. After each particle size determination test, the selected samples were packed in polyethylene packaging, vacuum packed, and stored until used in the extraction experiments. The average particle diameter was calculated using the methodology described by the American Society of Agricultural and Biological Engineers ASAE [36].

#### 2.2.2. True Density, Apparent Density, and Bed Porosity

The true density (*ρ*_r_) was performed using the helium gas pycnometry methodology, adopted due to its accuracy in determining the volume of the solid, with the appropriate specifications of the Pycnometer Accupyll 1340-Micromeritcs. The apparent density (*ρ*_a_) of the bed was determined by the ratio between the mass of the sample and the volume occupied in the extraction bed. The porosity of the bed (*ε*) was calculated by the relationship between the apparent density of the bed (*ρ*_a_) and the true density of the particles (*ρ*_r_), as shown in Equation (1):(1)$$\varepsilon =1-\frac{\rho_{a}}{\rho_{r}}$$

#### 2.2.3. Centesimal Composition

The centesimal composition of the lyophilized açaí pulp was performed according to the Association of Official Analytical Chemists [37]. The moisture contents were determined by the gravimetric method in an air circulation oven at 378.15 K. The ash contents were determined by incineration in a muffle furnace at 823.15 K. The total protein contents were obtained by the Kjeldahl method. The determination of total lipids was performed using Soxhlet extraction with petroleum ether solvent P.A. The determination of sugars was calculated by the Adolfo Lutz Institute method [38]. The determinations were made in triplicate and the results were expressed in grams per 100 g of sample on a dry basis (g/100 g d.b.).

### 2.3. Supercritical Extraction

The extraction with supercritical carbon dioxide was made aiming at the extraction of açaí oil and obtaining the pulp without fat. Therefore, the extraction tests were performed at the Extraction Laboratory of the Federal University of Pará (LABEX-UFPA), which has an appropriate infrastructure that consists of a unit Spe-ed SFE (Applied Separations, Inc., Allentown, PA, USA, model 7071) equipped with recirculating (POLYSCIENCE, F08400796, Niles, IL, USA), a compressor (SCHULZ, model CSA 7.8, Joinville, SC, Brazil), a flow meter at the system outlet (Alicat Scientific, M5SLPM, Tucson, AZ, USA), a cylinder of CO_2_ (Purity 99.9%, White Martins, Rio de Janeiro, RJ, Brazil), and a stainless steel extraction vessel suitable for operating at high pressures. The internal diameter of the vessel consists of an internal diameter of 0.01 m. The experimental extraction tests were performed under the following conditions of temperature and pressure: 323.15 K/35 MPa, 333.15 K/42 MPa, 343.15 K/490 MPa, according to Table 1. In these operating conditions, Batista et al. [32] obtained the best overall extraction yields, of which the density of carbon dioxide (CO_2_) is equal to or close to 900 kg/m^3^. After a previous selection of temperature and pressure conditions, the experimental extraction test was started with the preparation of the extraction bed, where an initial amount of 1.5 × 10^−4^ kg of lyophilized pulp was placed, with a supercritical solvent flow rate of 4 × 10^−5^ m^3^/s and a height of 0.06 m. Then, objectively and systematically, carbon dioxide was inserted into the system to contact the plant matrix and solubilize in açaí oil during the static period, which corresponded to the initial 900 s in relation to the total extraction time. Finally, after the static period, the açaí oil concentrated in bioactive compounds can be collected after the depressurization of supercritical carbon dioxide to ambient pressure and temperature. This oil collection step lasted 1800 s, which corresponded to the final extraction time, which is the dynamic extraction period. All experimental tests were performed in duplicate, the overall oil extraction yield was calculated in relation to the mass of oil extracted and the mass of lyophilized açaí pulp on a dry basis (d.b.) inserted at the beginning of extraction.

### 2.4. Characterization of Obtained Products

#### 2.4.1. Total Phenolic Compounds Content

The quantification of total phenolic compounds was defined for nonfat pulps, according to the Folin–Ciocalteu method [39]. The standard curve of gallic acid was made from 0.50 kg/m^3^ (*m*/*v*) stock solution, where fourteen points of concentration (0.83 to 17.25 mg/L) were used, according to equation y = 0.04 × –0.01 with R^2^ = 0.99. The concentration of the sample was equivalent to 52.63 kg/m^3^, taken to a refrigerated centrifuge for 1200 s at 133.3 rps, operating at 277.15 K. For the reactions, the concentration of the samples was equivalent to 0.4 (*v*/*v*). The readings were performed in a UV/VIS spectrophotometer (model UV-M90, Bel Engineering, Monza, Italy) at 760 nm. The determinations were performed in triplicate and the results were expressed in milligrams of gallic acid equivalent per gram of sample on a dry basis (mg GAE/g d.b.).

#### 2.4.2. Total Anthocyanins Content

The quantification of total anthocyanins was determined for the pulp without fat, using the same conditions operated for total phenolic compounds and the same centrifugation conditions. The compound was determined according to the differential spectrophotometric pH method described by Giusti and Wrolstad [40]. The absorbance readings were performed in a UV/VIS spectrophotometer (model UV-M90, Bel Engineering, Italy) at wavelengths of 510 and 700 nm. The determinations were performed in triplicate and the results were expressed in milligrams of 3-glycoside cyanide per 100 g of sample on a dry basis (3-glycoside cyanide/100 g d.b.).

#### 2.4.3. Carotenoids Content

The carotenoid contents were determined for açaí oils, according to the methodology of Rodriguez-Amaya and Kimura [41]. The petroleum ether P.A. was the solvent used to resuspend the oil. The readings were performed in a UV/VIS spectrophotometer (model UV-M90, Bel Engineering, Italy) at 450 nm. For the calculation, the absorptivity coefficient of β-carotene in petroleum ether (2592) was used. The determinations were made in triplicate and the results were expressed in micrograms of β-carotene per gram of sample in a dry base (µg/g d.b.).

#### 2.4.4. Antioxidant Activity

##### DPPH Method (2,2-diphenyl-1-picryl-hydrazyl)

Fat-free pulps and açaí oils were submitted to the determination of antioxidant activity by the spectrophotometric method DPPH (2,2-diphenyl-1-picril-hydrazil-hydrate), according to Brand-Williams et al. [42] with modifications, using the same proportion of the extraction solution operated for total phenolic compounds. The procedure was performed in a spectrophotometer (UV spectrophotometer/VIS (model UV-M90, Bel Engineering, Italy) at 515 nm, with readings monitored every 300 s until the reaction reached a plateau in 1800 s. The standard Trolox curve was performed from a 5 × 10^6^ µmol/m^3^ stock solution, where six concentrations (0 to 800 µM) were used according to the equation of a straight line y = 0.02x + 0.02 with R^2^ = 0.99. The reactions were performed using 5.00 × 10^−8^ m^3^ of sample and 1.95 × 10^−6^ m^3^ of DPPH solution at 0.06 mol/m^3^ (mol/v) (*v*/*v*). The antioxidant activity of the DPPH method was performed in triplicate and expressed in micromole of Trolox equivalent per gram of sample on a dry basis (µmol TE/g d.b).

##### ABTS Method (2,2′-azine-bis 3-ethylbenzoline-6-sulfonic Acid)

The ABTS method (2,2′-azine-bis acid (3-ethylbenzothiazoline-6-sulfonic) was carried out according to the procedure proposed by Miller et al. [43], and it was performed for both oils and nonfat pulps, preparing the 7 mM solutions of ABTS and 140 mM potassium persulfate, and consequently, its reaction for 600 s, in the absence of light. Thus, 1 × 10^−4^ m^3^ of this solution was diluted in ethyl alcohol until an absorbance reading of 0.70 ± 0.05 nm was found, using a spectrophotometer at 734 nm. The standard Trolox curve was performed at five concentrations (0–2000 μm), according to the equation of a straight line y = 3 × 10^−4^ x + 0.01 with R^2^ = 0.99. The experiment was carried out in triplicate, and the results were expressed in millimoles of Trolox equivalent per gram of sample on a dry basis (mmol Trolox eq./g d.b.).

##### Ferric Reducing Antioxidant Power Assay (FRAP)

To evaluate the antioxidant capacity by the FRAP method, for both oils and nonfat pulps, the methodology described by [44] was used. Initially, the FRAP reagent solution was prepared from the combination of 25 mL of 0.30 M sodium acetate buffer (pH 3.6), 2.5 mL of 10 nM TPTZ solution, and 2.5 mL of 20 nM ferric chloride aqueous solution. Subsequently, three dilutions were prepared at different concentrations of the extracts. An aliquot of 90 μL of each dilution was added to 270 μL of distilled water and 2.70 mL of FRAP reagent and homogenized and incubated for 1800 s at 37 C in a Maria bath (Fanem M100, São Paulo, SP, Brasil). After the incubation period, the reading was performed at 595 nm in a UV–visible absorption spectrophotometer, using the FRAP reagent as white. The results were expressed in μm of ferrous sulfate/g of the sample, calculated from the standard curve of ferrous sulfate.

### 2.5. Evaluation of Cytotoxic Activity

#### 2.5.1. In Vitro Cell Culture

The fat-free pulp and oil for the 323.15 K and 35 MPa extraction conditions were separated for cytotoxic evaluation with the two cell lines. The cell lines used in the study were cells (MRC-5) and (VERO). The cell lines used were MRC5 (ATCC: CCL-171™), derived from lung fibroblasts, and VERO (ATCC: CCL-81™), obtained from African green monkey kidneys. Both cell lines were procured from the American Type Culture Collection (ATCC) in Manassas, VA, USA. Cells were cultured in adherent monolayer cultures in Dulbecco’s Modified Eagle’s Medium (DMEM, Gibco^®^, Grand Island, NY, USA), supplemented with 10% fetal bovine serum and penicillin (100 U/mL) and streptomycin (100 mg/mL) (Gibco^®^), kept in an oven at 37 °C with an atmosphere of 5% CO_2_.

#### 2.5.2. MTT Essay

The cytotoxicity tests were performed by the MTT method, according to the literature [45,46], which consists of a colorimetric method based on the reduction of bromide salt of 3-(4,5-dimethylamazole-2-yl)-2,5-defenitetrazolio. The cells were seeded in 96-well plates at a density of 3 × 10^3^ cells/well for 86,400 s for adhesion to the plate. Subsequently, the cells were treated with the molecules under test at concentrations of 1.56–100 μg/mL for oil and 3.125–100 μg/mL for defatted pulp, and they were incubated at 310.15 K for 259,200 s. The analysis was performed in a plate spectrophotometer (SpectraMax i3, Molecular Devices, San Jose, CA, USA) at a wavelength of 570 nm. The tested substances were dissolved in dimethyl sulfoxide (DMSO) to obtain the stock concentration, and the experiments were performed in triplicate.

### 2.6. Statistical Analysis of Results

The mean and standard deviation were calculated for the characterization results and submitted to statistical analysis, with the aid of Statistica version 7.0, using the Tukey test with a significance level of 95% (*p* < 0.05). Cytotoxicity results were obtained in triplicate and submitted to analysis of variance (Oneway ANOVA), followed by the Bonferroni test, with a significance level of 95% (*p* < 0.05) in the GraphPad Prism Program, version 8.

## 3. Results and Discussion

### 3.1. Characterization and Pretreatment of Raw Material

The true average density of açaí samples was 1270 ± 0.02 (kg/m^3^). The apparent density of the samples was 143 kg/m^3^. The bed porosity was 0.9, and the mean particle diameter was 8.77 × 10^−4^ m. The result obtained for (dp) is acceptable for the processing of natural products according to the literature. According to Zabot et al. [47], the acceptable value is between 0.25 mm and 1.80 mm, and the appropriate particle diameter (dp) for the process can vary depending on the characteristics of the plant matrix and the diameter of the extraction vessel.

The result obtained by the analysis of the centesimal composition of lyophilized açaí pulp shows high concentrations of lipids, proteins, ash, moisture, and total sugars in amounts of 51.17 ± 0.00 g/100 g, 9.34 ± 0.88 g/100 g, 3.75 ± 0.12 g/100 g, 3.16 ± 0.14 g/100 g, and 21.21 ± 1.00 g/100 g, respectively. The high lipid content identified reflects a huge benefit for those who consume the fruit in natura, considering that this content has high concentrations of oleic acid (C18:0), which is beneficial for human health and can be used, even partially, in the prevention of cardiovascular diseases, as described by Cunha et al. [48]. Additionally, the protein content confers quality for the lyophilized pulp to be inserted as input in the formulation of high-protein cereal bars, which is of interest to the food industry [49]. Biocomposts of high antioxidant activity as total anthocyanins and total phenolic compounds, at concentrations of 693.85 ± 2.03 mg/100 g and 77.51 ± 0.54 mg GAE/100 g, were also identified in the lyophilized açaí pulp conferring quality for the açaí pulp to be a natural source of compounds of high antioxidant activity. The antioxidant capacity of the lyophilized açaí pulp was 289.08 ± 6.8 μmol TE/g, 385.92 ± 8.90 μmol TE/g, and 500.35 ± 8.66 μm of ferrous sulfate/g by the DPSO method, ABTS, and FRAP, respectively. All values were obtained on a dry basis, as shown in Table 1.

### 3.2. Determination of Global Income

The global oil extraction yields as a function of temperature and pressure operating conditions are shown in (Figure 1), where higher and lower extraction yield is observed in the condition of 343.15 K/49 MPa and 323.15 K/35 MPa. There was no statistically significant difference between the highest (51.74 ± 0.06%) and the lowest (50.14 ± 0.01%) yield. The density of CO_2_ in the experimental conditions used has a constant value of approximately 900 kg/m^3^, obtained from the NIST Chemistry Webbook (NIST Standard Reference Database). Density is a property responsible for the solvent capacity of supercritical fluids; increasing the density of the solvent increases its solvation power. But in constant density, little influence of temperature and pressure on the overall yield was expected. Similar results were obtained by Silva et al. [33] and Batista et al. [32], who obtained extraction yields of açaí oil of 57.06 ± 0.03% and 40.81% under the same pressure and temperature conditions considered in this study. However, it is worth mentioning that although Silva et al. [33] and Batista et al. [32] used açaí of the species *Euterpe oleraceae*, the difference in extraction yields may have been influenced by external factors associated with the collection site, planting condition, degree of ripeness, and period of harvest of the fruit [50].

In a comparative analysis with other authors who applied supercritical carbon dioxide technology to obtain vegetable oil from typical Amazon palm fruits, such as *Endopleura uchi* [51], *Oenocarpus distichus Mart.* [48], *Byrsonima crassifolia* [52], and *Astrocaryum vulgare* [53], with yields of 24.48 ± 0.20%, 45.90 ± 0.15%, 9.12 ± 0.08%, and 31.49 ± 0.04%, the efficiency of supercritical carbon dioxide in solubilization and extraction of açaí oil had a yield of 50.14 ± 0.01%. This result shows the efficiency of supercritical technology with carbon dioxide as a solvent for extraction of açaí oil, if the almost integral removal of the lipids initially identified in the centesimal composition is considered. After the extraction of açaí oil with supercritical carbon dioxide, the pulp without fat was obtained.

The fatty acid profiles of açaí oils obtained in this work by CO_2_-SE under 323.15 K/35 MPa, 333.15 K/42 MPa, and 343.15 K/49 MPa conditions were not made. According to the literature, there is no significant difference in the content of fatty acids for the operating conditions studied. The effect on the fatty acid composition of açaí oil was more accentuated by geographic location than by the operating conditions of CO_2_-SE [33].

### 3.3. Phenolic Compounds Content

The total content of phenolic compounds in the pulp without fat was concentrated in relation to lyophilized pulp (77.51 ± 0.54) mg GAE/g d.b. after supercritical extraction, ranging from 150.20 ± 1.32 to 96.43 ± 0.46 mg GAE/g d.b., according to Table 1. It was also observed that there was significant variation in the conditions analyzed, whose highest content of phenolic compounds is in the lowest condition of temperature and pressure (323.15 K to 35 MPa), and the lowest content is for 343.15 K to 49 MPa. With the increase in pressure and temperature, we observed two effects: (a) an increase in pressure decreased the levels of phenolic compounds, possibly due to the fact that the increase in pressure facilitated the enzymatic oxidation of the fat-free pulps [54]; (b) with an increase in temperature, there was a degradation of thermosensitive compounds. This behavior was also observed by Batista et al. [26] for açaí pulp without fat, obtained under the same extraction conditions as in the present study. In addition, the levels of phenolic compounds in this work are higher than those found for fat-free pulps of Murici (Byrsonima crassifolia) at 22 MPa and 695 kg/m^3^ of supercritical CO_2_ (24.58 ± 0.86) mg GAE/g d.b. [55], as well as for black cumin seeds without fat (*Nigella sativa* L.) (35.6 mg GAE/g d.b.) [56]. Therefore, the obtained product can be marketed not only on account of the class of anthocyanins, a major class of phenolic compounds, but also by other compounds, known as nonanthocyanins, for their antioxidant (3.4 dihydroxybenzoic, p-dihydroxybenzoic, isoferulic, syringaresinol, p-coumaric acids) [57,58,59,60], antimicrobial (vanilic acid) [61], therapeutic (caffeic acid) [62], and anti-inflammatory effects (isoorientin and orientin) [63]. It is noteworthy that the commercialization of the pulp without the fat of açai can cover the cosmetics and food industries, as it has already been applied in formulations for other vegetable matrices, such as grape (*Vitis vinífera*), pomegranate (*Punica Granatum*), green tea (*Camellia sinensis*) and Pycnogenol (substance extracted from French pine bark) [64,65]. The pharmaceutical industry also forms another branch of application of the obtained product [64], since there are studies suggesting the extraction of these compounds from other raw materials for investment in this field, such as the extraction of phenolic compounds from coffee byproducts [66].

### 3.4. Total Anthocyanins Content

The total content of anthocyanins in the pulp without fat was concentrated in relation to lyophilized pulp ((693.85 ± 2.03) mg/100 g d.b.) after extraction under different operating conditions, ranging from 1330.50 ± 11.05 to 1384.39 ± 64.72 mg/100 g, according to Table 1, and there was no significant difference between the conditions analyzed, which indicates that the increase in temperature and pressure did not influence the content of anthocyanins. This behavior was also observed in some localities by Silva et al. [33]; much higher values were found in this study compared with those found by Batista et al. [32] for fat-free açaí pulp obtained under the same extraction conditions as the present study. The difference in anthocyanin content may have been influenced by external factors associated with the collection site, planting conditions, degree of ripeness, and harvest period of the fruit [50]. Aiming at the industrial application of this product, these levels of anthocyanins found in pulps without açaí fat indicate a great potential for applicability in nutraceuticals [67], because its concentration in this work is higher than the main raw materials considered a source of anthocyanins, such as mulberries, black chokeberries, black elderberries, blackberries, strawberries, black grapes, and blueberries, among others, ranging from 1 to 704 mg/100 g [68]. Still, in the food field, this application would influence the formulation of antioxidant supplements, as well as the use of pigments of natural origin, such as dyes, due to the specific characteristics of this compound [69]. In addition, it has the ability to eliminate free radicals and acts in a series of enzymes such as cyclooxygenase- and mitogen-activated protein kinase, and in the signaling of inflammatory cytokines [70], it has also been shown that there is evidence for the prevention of and support to combat diseases, which indicates possible pharmacological applications [3]. The cosmetics industry is also another target of applicability of this product due to levels significantly higher than those already found in the market, which already uses this compound from the purple sweet potato in the formulation of a cosmetic product at the concentration of 0.61 mg/100 g of cream [71]. Thus, the high applicability of the pulp without açaí fat in the market is deduced due to its physical, chemical, and biological properties.

### 3.5. Total Carotenoids Content

The carotenoid content of açaí oils could be expressed after supercritical extraction under different operating conditions, ranging from 250.40 ± 2.10 µg/g d.b. to 277.09 ± 3.65 µg/g d.b., as shown in Table 1, and there was significant influence on the conditions analyzed. The lowest carotenoid content was obtained at temperature and pressure conditions of 333.15 K to 42 MPa, and the highest value was at the highest temperature and pressure conditions (343.15 K to 49 MPa). The highest carotenoid content in the highest condition was also observed by Pires et al. [55] for Murici oil, obtained in the same extraction condition as the present study. In addition, the values found in this study were higher than those found by da Cunha et al. [48] for the oil extracted by cold and hot pressing of the *Euterpe edulis* palm., as well as those found by Santos, Alves, and Roca for the oil extracted by the traditional Soxhlet method of the *Attalea maripa* palm [56]. It is also important to note that these compounds are known due to their orange, red, and yellow coloration [72], and thus are exploited in raw materials such as buriti [73], tucumã [74], and pupunha [75]. However, although açaí has purple or black coloration, it has the particularity of its oil having colors ranging from yellow to green, which indicates that the fruit is a source of carotenoids and that the oil obtained can be commercialized industrially due to its antioxidant action and photosensitivity protection [17]. Because of these characteristics, in the food industry, the consumption of carotenoids implies the mechanical strength of cells, and the oil of this fruit can be used as olive oil, since it approximates the chemical composition of olive oil that is already widely marketed due to a voluminous amount of monounsaturated and polyunsaturated fatty acids [5,76,77]. In addition, according to Duncan [78], the carotenoid with the highest percentage found in açaí oil is β-carotene, a compound that has been indicated to have a wide range of cosmetic applications, since studies suggest that the topical use of β-carotene in mice and guinea-pigs (India) revealed protective properties against UVA radiation [79]. In the pharmacological industry, β-carotene has also been used in vitamin supplements and oral minerals for prenatal/postpartum conditions, as it influences the reduction in photosensitivity in patients with erythropoietic protoporphyria and other photosensitivity diseases [80], which demonstrates one more branch of application of the product obtained. The usefulness of carotenoids has been associated with nutri cosmetic applications [81], that is, in the oral consumption of products containing food components for cosmetic purposes. In this context, nineteen nutri cosmetics were analyzed in the literature, and in general, these supplements contain between 0.01 and 18.22 mg [82]. This shows, once again, that açaí oil can be placed on the market as a new source of carotenoids due to the content of this bioactive compound in the product.

### 3.6. DPPH ASSAY

The antioxidant activity of the nonfat pulps for the DPPH method was concentrated from the lyophilized pulp (289.08 ± 6.83) μmol TE/g d.b. after extraction under different operating conditions, ranging from 414.99 ± 5.02 μmol TE/g d.b. to 362.71 ± 71.07 μmol TE/g d.b., according to Table 1. For nonfat pulps, it is emphasized that there was significant variation for this test, whose highest value was obtained in the lowest temperature and pressure condition (323.15 K to 35 MPa) and the lowest value in the highest temperature and pressure condition (343.15 K to 49 MPa). It can be observed that the increase in temperature and pressure decreased the antioxidant activity of the defatted pulp, and this behavior was also observed by Pires et al. [55] for defatted Murici pulp, obtained under the same extraction conditions of the present study. The results found in this work are lower than those found in the literature (707.56 ± 19.12) µmol Trolox/g [33], and this can be explained by the difference in the collection site, planting condition, degree of maturation, and fruit harvest period [50]. Likewise, despite different values, these results configure great antioxidant properties of the pulp without fat. The decrease in antioxidant activity with increasing temperature and pressure of the operating conditions studied indicates the same behavior of phenolic compounds, so these compounds were the main contributors to antioxidant activity by this method. The same compounds have been reported as the main contributors to the high antioxidant activity of açaí [83]. The behavior between the DPPH and FRAP methods is the same, and this can be explained by the identical mechanisms of action of the methods, which involve electron transfer [84].

For the açaí oil obtained, the antioxidant activity for this method ranged from 2.02 ± 0.07 μmol TE/g d.b. to 2.55 ± 0.14 μmol TE/g d.b., according to Table 1, and there was a significant difference: the lowest value obtained was in the lowest condition of temperature and pressure (323.15 K to 35 MPa) and the highest value in the highest temperature and pressure condition (343.15 K to 49 MPa). The results found in this work are lower than those reported by Pires et al. [55] for Murici oil. It was observed that the antioxidant activity exhibited the same behavior as in the ABTS assays, as there was an increase in this activity with the rise in temperature and pressure. This result can be explained by the methods involving electron transfer reactions [42,85,86]. Thus, both products obtained by supercritical technology showed satisfactory values by this method, which allows one to say that they are good sources of antioxidant compounds.

### 3.7. ABTS ASSAY

The antioxidant activity of the fat-free pulp for the ABTS method was concentrated from the lyophilized pulp (385.92 ± 8.90) μmol TE/g d.b. after extraction with supercritical CO_2_ under the different operating conditions, ranging from 554.53 ± 7.68 μmol TE/g d.b. to 23.644 μmol TE/g d.b., as shown in Table 1. It was observed that there was a significant influence on the conditions used for the antioxidant activity of this method, whose lowest value was obtained in the lowest condition of temperature and pressure (323.15 K to 35 MPa) and whose highest value was in the highest condition of temperature and pressure (343.15 K to 49 MPa). The results found in this work are superior to those found in the literature, ranging from 9.10 ± 0.40) μmol Trolox/g to 9.40 ± 0.20 μmol Trolox/g [87]. Regarding the nonfat pulps, it was observed that the antioxidant activity had a distinct behavior of phenolic compounds, total anthocyanins, and DPPH assays, since there was an increase in this activity with increasing temperature and pressure. According to the literature, although this method also involves electron transfer reactions, there are other possible explanations for why the same samples do not yield the same results across all methodologies. This could be due to different characteristics and mechanisms of action of the bioactive compounds in the samples and the reagents [42,85,86,88].

For açaí oils, the antioxidant activity for this method ranged from 2.05 ± 0.04 μmol TE/g d.b. to 2.60 ± 0.05 μmol TE/g d.b., according to Table 1, and there was a significant difference. In addition, the value achieved in this study for antioxidant activity by the ABTS method is higher than that found by Pinto et al. [87] for Bacaba oil extracted in the same working extraction conditions (333.14 K to 42 MPa). It was observed that the antioxidant activity exhibited the same behavior as the DPPH assays, as there was an increase in this activity with the rise in temperature and pressure. This result can be explained by the methods involving electron transfer reactions [42,85,86]. Thus, both products obtained by supercritical technology showed satisfactory values by this method, indicating excellent sources of antioxidants.

### 3.8. FRAP METHOD

The antioxidant activity of the nonfat pulps for the FRAP method was concentrated from the lyophilized pulp (500.35 ± 8.66) μm of ferrous sulfate/g d.b.) after extraction under different operating conditions, ranging from 746.25 ± 3.82 μm of ferrous sulfate/d.b. to 653.13 ± 9.97 μm of ferrous sulfate/g d.b., according to Table 1. For nonfat pulps, it is emphasized that there was significant variation for this test, whose highest value was obtained in the lowest temperature and pressure condition (323.15 K to 35 MPa) and the lowest value in the highest temperature and pressure condition (343.15 K to 49 MPa). The antioxidant activity of many fruits arises from the combined synergistic action of a mixture of compounds, including phenolic compounds, carotenoids, and vitamins C and E. However, in fruits such as açai, which contain relatively small amounts of these vitamins, phenolic acids, flavonoids, and anthocyanins are the main contributors to antioxidant activity [89,90]. The decrease in antioxidant activity measured by the reducing capacity of Fe^3+^ with the increase in temperature and pressure of the operating conditions studied indicates a strong correlation and the same behavior of phenolic compounds, so these compounds were the main contributors to antioxidant activity. The same behavior was observed for Pataua in the study by Hidalgo et al. [86]. The behavior between the DPPH and FRAP methods is the same, and this is explained by the mechanisms of action of the equal methods and electron transfer [84].

For the açaí oil obtained, the antioxidant activity for this method varied from 9.95 ± 0.89 μmol TE/g d.b. to 15.25 ± 1.69 μmol TE/g d.b., according to Table 1, and there was a significant difference: the highest value obtained was under the condition of 343.15 K and 49 MPa pressure. It was observed that the antioxidant activity did not exhibit the same behavior in the ABTS and DPPH assays. According to the literature, despite this method also involving electron transfer reactions, there are other possible explanations for why the same samples do not yield the same results across all methodologies. These differences could be due to the varying characteristics and mechanisms of action of the bioactive compounds in the samples and the reagents [42,85,86,88]. Thus, both products obtained by supercritical technology showed satisfactory values by this method, which allows one to say that they are good sources of antioxidants.

### 3.9. In Vitro Evaluation of Cytotoxicity

For the development of food products, cosmetics, and pharmaceuticals, the evaluation of cytotoxicity is important to evaluate the use concentrations of bioactive compounds. For this cytotoxicity test, the oil and fat-free pulps were used for the best extraction condition (323.15 K to 35 MPa), chosen according to the methodology previously presented. The data for the cytotoxicity tests for the oil are shown in Figure 2 and for the fat-free pulp in Figure 3. According to Figure 2, the percentage of cell viability varied from 96 to 100% for MRC-5 and from 100 to 103% for VERO. It was observed that there was no significant variation in cell viability, so the extracted açaí oil in the condition analyzed by this method did not present cytotoxicity on the normal cell lines tested. Regarding pulp without açaí fat (Figure 3), the percentage of cell viability ranged from 86 to 99% for MRC-5 cells and 88 to 102% for VERO cells. This difference in the percentage of viability can be explained according to recent studies, which assume that MTT assays can interfere with the cell culture medium [91]. However, the results show that even at higher concentrations of the pulp without fat, cell viability remains very similar to the control without treatment, close to 100%. Unlike the literature that, with the use of ethanol as solvent for the extraction of anthocyanins, demonstrated cytotoxicity in certain concentrations of crude and semipurified blackberry extracts [92]. Supercritical extraction is, therefore, an ideal technology for obtaining nontoxic products, which has already been studied according to the literature [93,94,95]. In this sense, the data indicate that oil and pulp without açaí fat, under the condition of extraction with supercritical CO_2_ analyzed and in this concentration content, do not produce effects on MRC-5 and VERO cells, which may indicate the safe use of these products in future applications, such as in the areas of cosmetics, pharmaceuticals, and food.

## 4. Conclusions

The açaí oils showed excellent levels of yield, carotenoids, and antioxidant activity, as well as the nonfat pulps, with high levels of anthocyanins, phenolic compounds, and antioxidant activity. Thus, we obtained products with nutritional quality that can be applied in various industrial branches.

The anthocyanin content, in any supercritical extraction condition reported, showed no significant difference.

The cytotoxicity tests did not exhibit cytotoxic activity in the normal cell lines tested (MRC-5 and VERO) for the samples analyzed in the operational condition 323.15 K and 35 MPa, showed no significant differences in cell viability with increased concentration of both products.

The results of this study suggest pharmacological, food, and cosmetic potentials in the commercialization of açaí products obtained by the extraction with supercritical CO_2_, due to the selectivity of the technology. These products can be considered innovative as they come from green and sustainable technology, contribute to the preservation of the environment, and boost the bioeconomy, ratifying the fruit to a global level.

## Figures and Tables

**Figure 1 foods-13-02819-f001:**
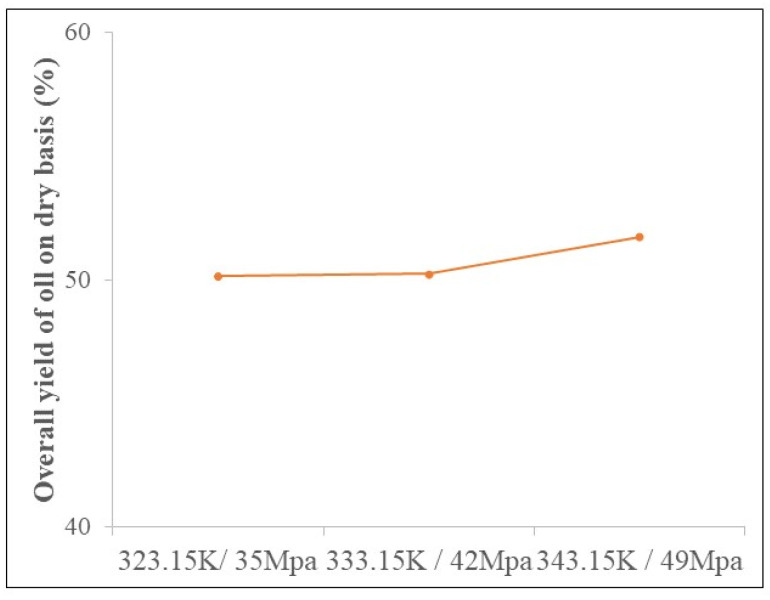
Global yield of açaí pulp oil (*Euterpe oleracea*) obtained by CO_2_-SFE (standard deviations ≤ 0.9%).

**Figure 2 foods-13-02819-f002:**
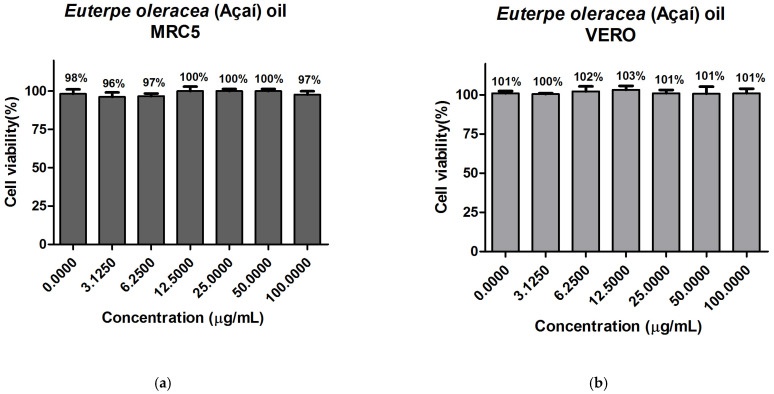
Graph of the effect of Euterpe oleracea (Açaí) oil obtained at 323.15 K and 35 MPa by supercritical CO_2_: (**a**) MRC-5 cell viability; (**b**) VERO cell viability; CN (negative control).

**Figure 3 foods-13-02819-f003:**
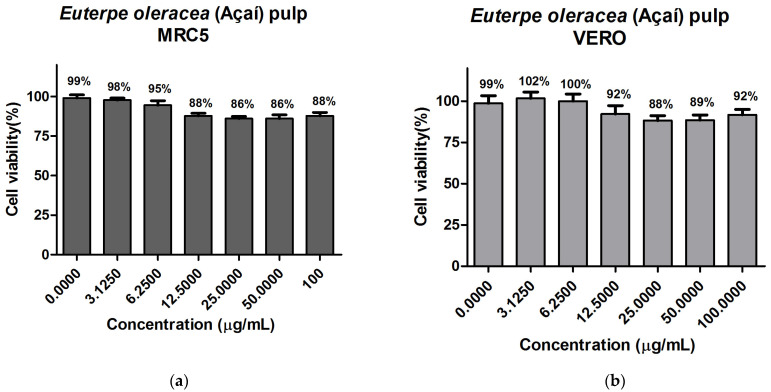
Graph of the effect of Euterpe oleracea (Açaí) pulp obtained at 323.15 K and 35 MPa by supercritical CO_2_: (**a**) MRC-5 cell viability; (**b**) VERO cell viability; CN (negative control).

**Table 1 foods-13-02819-t001:** Operating conditions of CO_2_-SFE açaí pulp (*Euterpe oleracea*) and characterization of the different defatted pulps and oils.

Sample	T (K)	P (MPa)	Yield (% d.b.)	Phenolic Compounds (mg GAE/g d.b.)	Anthocyanin (mg/100 g d.b.)	Carotenoids (µg/g d.b.)	DPPH (μmol TE/g d.b.)	ABTS (μmol TE/g d.b.)	FRAP (Ferrous Sulfate μm/g d.b.)
Lyophilized Pulp	-	-	-	77.51 ± 0.54 ^d^	693.85 ± 2.03 ^b^	-	289.08 ± 6.83 ^d^	385.92 ± 8.90 ^d^	500.35 ± 8.66 ^c^
Fat-free Pulp	323.15	35	n.d	150.20 ± 1.32 ^a^	1330.50 ± 11.05 ^a^	n.d	414.99 ± 5.02 ^a^	554.53 ± 7.68 ^c^	746.25 ± 3.82 ^a^
	333.15	42	n.d	109.14 ± 0.72 ^b^	1324.82 ± 55.44 ^a^	n.d	388.93 ± 4.65 ^b^	589.74 ± 0.00 ^b^	726.15 ± 6.79 ^a^
	343.15	49	n.d	96.43 ± 0.46 ^c^	1384.39 ± 64.72 ^a^	n.d	362.71 ± 5.07 ^c^	644.23 ± 7.23 ^a^	653.13 ± 9.97 ^b^
Oils	323.15	35	50.14 ± 0.01 ^b^	n.d	n.d	250.40 ± 2.10 ^b^	2.02 ± 0.07 ^b^	2.05 ± 0.04 ^c^	9.95 ± 0.89 ^b^
	333.15	42	50.24 ± 0.97 ^b^	n.d	n.d	246.22 ± 2.51 ^c^	2.40 ± 0.11 ^a^	2.24 ± 0.11 ^b^	5.50 ± 9.97 ^c^
	343.15	49	51.74 ± 0.06 ^a^	n.d	n.d	277.09 ± 3.65 ^a^	2.55 ± 0.14 ^a^	2.60 ± 0.05 ^a^	15.25 ± 1.69 ^a^

Different letters in the same column, per sample, show a difference in significance level of 5% (*p* < 0.05); n.d: not detected.

## Data Availability

The original contributions presented in the study are included in the article, further inquiries can be directed to the corresponding author.

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
