# Peer review of "Evaluation of Bioactive Compounds and Antioxidant and Cytotoxic Effects of Oil and Pulp without Açaí Fat (Euterpe oleracea) Obtained by Supercritical Extraction"

_foods, 2024, doi:10.3390/foods13172819_

Round 1

Reviewer 1 Report

Comments and Suggestions for Authors

I have reviewed the paper entitled “Evaluation of Bioactive Compounds and Antioxidant and Cyto-toxic Effects of oil and pulp without açai fat (Euterpe Oleracea) obtained by Supercritical Extraction” and recommend its acceptance subject to significant revisions. The specific recommendations are as follows:

1. It is recommended that the importance of this study be emphasized at the end of the abstract.

2. Keywords are suggested to be modified to add the research object of this study.

3. The introductory section should provide more details on the biological activities and research background of Euterpe oleracea and the advantages of supercritical CO2 extraction technology.

4. It is recommended that the literature review section be expanded to cover more relevant studies to demonstrate the relevance and innovation of this study to the existing literature.

5. A more detailed description of the origin and handling of the experimental material is needed, such as the maturity of the açaí fruits and the collection conditions, which may influence the results of the experiment.

6. It is recommended that the tests for the determination of physical and chemical properties be supported by additional relevant literature.

7. Check graphs to ensure consistency of text formatting in graphs.

8. Trends in research findings should be discussed in depth in terms of the mechanisms of change, e.g., as in lines 477-482, and supported by documentation.

9. Line 499, provide references.

10. The authors mentioned several times that there is a correlation between phenolics, carotenoids and antioxidant activity, so it is suggested that additional correlation analyses are supported.

11. The results section should present the data in more detail, including detailed data for all experimental conditions, not just the optimal ones. Also, the results need to be analyzed and interpreted in more depth, especially the effect of different conditions on the content of bioactive compounds.

12. The discussion section should explore in more depth the significance of the experimental results, including comparisons with other studies.

13. The conclusion section should be based on the experimental results and the content of the discussion section, avoiding excessive extrapolation or making conclusions that are not supported by the data.

Author Response

  1. It is recommended that the importance of this study be emphasized at the end of the abstract.

Response 1: Thank you for suggestions and commentary. The text was corrected on abstract in lines 47-48.

  1. Keywords are suggested to be modified to add the research object of this study.

Response 2: Thank you for suggestions and commentary. The text was corrected on keywords in lines 49.

  1. The introductory section should provide more details on the biological activities and research background of Euterpe oleraceaand the advantages of supercritical CO2 extraction technology.

Response 3: Thank you for your comment. Done as suggested in lines 70, 77-80 and 98-102.

  1. It is recommended that the literature review section be expanded to cover more relevant studies to demonstrate the relevance and innovation of this study to the existing literature.

Response 4: Thank you for suggestions and commentary. Done as suggested in lines 109-114.

  1. A more detailed description of the origin and handling of the experimental material is needed, such as the maturity of the açaí fruits and the collection conditions, which may influence the results of the experiment.

Response 5: Thank you for suggestions and commentary. Done as suggested in lines 122-123.

  1. It is recommended that the tests for the determination of physical and chemical properties be supported by additional relevant literature.

Response 6: Thank you for suggestions and commentary. Done as suggested in lines 145 and 243-244.

  1. Check graphs to ensure consistency of text formatting in graphs.

Response 7: Thank you for suggestions and commentary. The figures have been revised.

  1. Trends in research findings should be discussed in depth in terms of the mechanisms of change, e.g., as in lines 477-482, and supported by documentation.

Response 8: Thank you for suggestions and commentary. They were added in lines 367-371, 473-476, 481-485, 501-505, 510-513, 532-533, 540-544, and supported by the literature.

9- Line 499, provide references.

Response 9: Thank you for suggestions and commentary. That line was removed and new references were added in lines 510-513.

10- The authors mentioned several times that there is a correlation between phenolics, carotenoids and antioxidant activity, so it is suggested that additional correlation analyses are supported.

Response 10: Thank you for suggestions and commentary. New additional correlation analyses have been added in lines 473-474 and 532-533.

11- The results section should present the data in more detail, including detailed data for all experimental conditions, not just the optimal ones. Also, the results need to be analyzed and interpreted in more depth, especially the effect of different conditions on the content of bioactive compounds.

Response 11: Thank you for suggestions and commentary. The objective of this work was not to detail all the conditions, but rather to present their general behavior. The results were further analyzed and included in lines 367-371 and 397-402.

12- The discussion section should explore in more depth the significance of the experimental results, including comparisons with other studies.

Response 12: Thank you for suggestions and commentary. More comparisons with other studies were included in the lines. 367-371, 397-402, 473-476, 481-485, 501-505, 510-513, 532-533 e 540-544.

  1. The conclusion section should be based on the experimental results and the content of the discussion section, avoiding excessive extrapolation or making conclusions that are not supported by the data.

Response 13: Thank you for suggestions and commentary. The conclusions were reformulated.

Reviewer 2 Report

Comments and Suggestions for Authors

In this paper, the author aim to obtain the bioactive of oil and pulp of açaí fat-free supercritical technology and evaluate the cytotoxicity of products in MRC-5 and VERO cells in vitro. Overall, this work is interesting. However, some major revisions should be required before acceptance in the manuscript:

1. Both temperature and pressure affect the yield, however, the author does not seem to use a single factor analysis, whether the conditions are set reasonably?

2. Lines 355-377, The authors attributed the decrease in levels of phenolic compounds to the increase in pressure, however, the factor of temperature was not considered.

3. Line 550, 2 in “CO2” should be the subscript.

4. The fonts in the Figures should be consistent.

Comments on the Quality of English Language

Please check the grammar carefully, such as Line 90-94, line 352-355, etc..

Author Response

  1. Comments 1: Both temperature and pressure affect the yield, however, the author does not seem to use a single factor analysis, whether the conditions are set reasonably?

Response 1: The variation in temperature and pressure affects the density of CO2 and consequently the solubility capacity of the solute. However, at constant density, little influence of temperature and pressure on the overall yield was expected, as discussed in lines 318-327. These conditions were chosen according to the best yields obtained in previous studies. Batista et al. [32] and Silva et al. [33].

32. Batista, C. De C. R.; De Oliveira, M. S.; Araújo, M.E.; Rodrigues, A.M.C.; Botelho, J.R.S.; Souza, A.P. Da S.; Machado, N.T.; De Carvalho, R.N. Supercritical CO2 extraction of açaí (Euterpe oleracea) berry oil: Global yield, fatty acids, allelopathic activities, and determination of phenolic and anthocyanins total compounds in the residual pulp, Supercrit. Fluids 2016, 107, 364–369. https://doi.org/10.1016/J.SUPFLU.2015.10.006.

33. Silva, M.P.; Cunha, V.M.B.; Sousa, S.H.B.; Menezes, E.G.O.; Bezerra, P.D.N.; e Farias Neto, J.T.; Filho, G.N.R.; Araújo, M.E.; De Carvalho, R.N. Supercritical CO2 extraction of lyophilized Açaí (Euterpe oleracea Mart.) pulp oil from three municipalities in the state of Pará, Brazil, CO2 Util. 2019, 31, 226–234. https://doi.org/10.1016/J.JCOU.2019.03.019.

  1. Comments 2:  Lines 355-377, The authors attributed the decrease in levels of phenolic compounds to the increase in pressure, however, the factor of temperature was not considered.

Response 2: Thank you for your opinion. The new text was inserted in lines 367-371.

  1. Comments 3: Line 550, 2 in “CO2” should be the subscript.

Response 3: Thank you for your opinion. Lines 550 It was rewritten in line 580.

  1. Comments 4: The fonts in the Figures should be consistent.

Response 4: Thank you for your opinion. The figures used in this work are the results of the authors' experiments and are supported by scientific literature in the areas of supercritical fluid extraction and cytotoxicity.

Reviewer 3 Report

Comments and Suggestions for Authors

This article is interesting, however some comments are made.

Line 112. Wouldn't it be better to write 15 hours? The same comment in lines 118, 126, 171, 174, 187 (in min), 220.

Lines 237-247. Please eliminate this paragraph because is repeated in the next section.

More discussion is needed in this section “3.1. Characterization and pre-treatment of raw material”

Much discussion of the structure-activity relationship is missing from all analyzes. Even though the individual identification of the compounds has not been carried out, there is already information in the references of the majority compounds or of the structures in general. Likewise, more discussion is needed on the mechanism or form of interaction of CO2 and the pressure on metabolites and their activities.

Additionally, there is a lack of discussion on the antioxidant mechanisms of each method, affinity of the compounds for each method, etc.

In the cytotoxicity part, don't you have the average cytotoxic concentration or something similar? so that you have an idea of ​​how long the extract may be toxic.

The first part of the conclusions seems more like a summary of the results. The authors could restructure it.

Author Response

  1. Comments 1: Line 112. Wouldn't it be better to write 15 hours? The same comment in lines 118, 126, 171, 174, 187 (in min), 220.

 Response 1: Thank you for your comment. The text in line 112 has been revised. The authors use the International System of Units (SI) where time is given in seconds.

  1. Comments 2: Lines 237-247. Please eliminate this paragraph because is repeated in the next section.

 Response 2: Thank you for your comment, Lines 237-247 have been removed.

  1. Comments 3: More discussion is needed in this section “3.1. Characterization and pre-treatment of raw material”

Response 3: Thank you for your comment, more discussions have been added in lines 301-305.

  1. Comments 4: Much discussion of the structure-activity relationship is missing from all analyzes. Even though the individual identification of the compounds has not been carried out, there is already information in the references of the majority compounds or of the structures in general. Likewise, more discussion is needed on the mechanism or form of interaction of CO2and the pressure on metabolites and their activities.

Response 4: Thank you for your opinion. Discussing the structure-activity relationship of the compounds was not the objective of this work, and further discussions were added in the lines. 367-371, 473-476, 481-485, 501-505, 510-513, 532-533, 540-544.

  1. Comments 5: Additionally, there is a lack of discussion on the antioxidant mechanisms of each method, affinity of the compounds for each method, etc.

Response 5: Thank you for your opinion. New discussions on the antioxidant mechanisms of each method have been added in the lines 473-476, 481-485, 501-505, 510-513, 532-533, 540-544.

  1. Comments 6: In the cytotoxicity part, don't you have the average cytotoxic concentration or something similar? so that you have an idea of ​​how long the extract may be toxic.

Response 6: We appreciate your careful revision of our manuscript. We employed a concentration curve from 3.125 to 100 µg/mL in the cytotoxicity experiments. However, we could not determine the mean cytotoxic concentration (IC50) because there was no significant difference in the cell viability among the treatment groups. This point indicates that the substance does not significantly influence the cell viability up to 100 µg/mL, and its IC50 must lay above this value. We are confident in our findings and believe that this information provides insights into the substance's effects on the cell viability of normal cells.

  1. Comments 7: The first part of the conclusions seems more like a summary of the results. The authors could restructure it.

Response 7: Thank you for your opinion. The conclusions were reformulated.